# Time-Encoding-Based Ultra-Low Power Features Extraction Circuit for Speech Recognition Tasks

**Eric Gutierrez \*** , **Carlos Perez** , **Fernando Hernandez and Luis Hernandez**

Department of Electronics Technology, Carlos III University of Madrid, 28911 Leganes, Spain; caperezc@ing.uc3m.es (C.P.); ferherna@ing.uc3m.es (F.H.); luish@ing.uc3m.es (L.H.)

**\*** Correspondence: eric.gutierrez@uc3m.es; Tel.: +34-916245987

**Abstract:** Current trends towards on-edge computing on smart portable devices requires ultra-low power circuits to be able to make feature extraction and classification tasks of patterns. This manuscript proposes a novel approach for feature extraction operations in speech recognition/voice activity detection tasks suitable for portable devices. Whereas conventional approaches are based on either completely analog or digital structures, we propose a "hybrid" approach by means of voltage-controlled-oscillators. Our proposal makes use of a bank a band-pass filters implemented with ring-oscillators to extract the features (energy within different frequency bands) of input audio signals and digitize them. Afterwards, these data will input a digital classification stage such as a neural network. Ring-oscillators are structures with a digital nature, which makes them highly scalable with the possibility of designing them with minimum length devices. Additionally, due to their inherent phase integration, low-frequency band-pass filters can be implemented without large capacitors. Consequently, we strongly benefit from power consumption and area savings. Finally, our proposal may incorporate the analog-to-digital converter into the structure of the own features extractor circuit to make the full conversion of the raw data when triggered. This supposes a unique advantage with respect to other approaches. The architecture is described and proposed at system-level, along with behavioral simulations made to check whether the performance is the expected one or not. Then the structure is designed with a 65-nm CMOS process to estimate the power consumption and area on a silicon implementation. The results show that our solution is very promising in terms of occupied area with a competitive power consumption in comparison to other state-of-the-art solutions.

**Keywords:** artificial intelligence; machine learning; speech recognition; features extraction; voltage-controlled-oscillator; analog-to-digital converter

## 1. Introduction

High computing capability of portable devices has made possible the implementation over them of voice user interfaces such as speech recognition or keyword spotting [1,2]. Nevertheless, conventional digital processing of the microphone input cannot be made uninterruptedly due to power limitations [3]. One possible solution consists of making the processing on the cloud. However, this may suppose issues related to user privacy or latency. Consequently, the trend is towards an on-edge computing with ultra-low power architectures [4,5]. In relation to these topics, Voice-Activity-Detectors (VADs) have become of interest in the last years with the goal of detecting if an audio input stream is a human voice or environmental noise [6–8].

Looking at VAD or speech recognition applications, we may distinguish between two approaches, the digital approach and the analog one. The first approach consists of turning the input analog raw data into digital data and making intensive digital computing (windowing, FFT, filter operations and

power calculations (Figure 1a) to estimate the features required to detect data patterns. Afterwards, a classification stage, such as a feed-forward neural network or a decision tree, decides whether the data correspond to the human voice or not. The use of this architecture in portable devices is restricted by the power consumption of digital circuits, which may need high-capacity batteries [9].

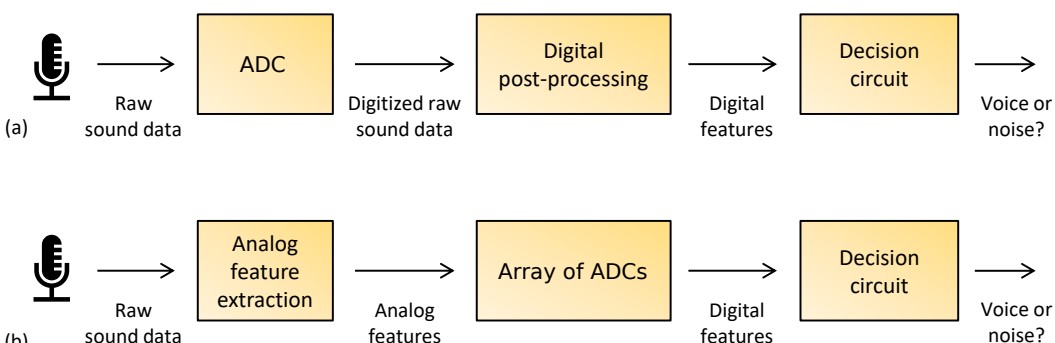

**Figure 1.** Speech recognition/Voice-Activity-Detector (VAD) approaches: (**a**) digital features extraction; and (**b**) analog features extraction.

Because of this battery-life limitation, a second approach have been recently proposed. It is based on making equivalent features extraction operations (previously digitally made) but in the analog domain, and then performing the analog-to-digital conversion (Figure 1b). To implement the analog features extraction, a bank of band-pass filters and power estimators are used in order to get the energy of the input signal filtered within different bands of interest. The number of filters in the bank sets the frequency resolution. Then, a classification circuit distinguishes between noise and voice. The advantage of the analog approach is the enormous reduction of the power consumption in the features extractor circuit, making them suitable for ultra-low power applications [8]. However, these architectures often make use of large capacitors due to the low-frequency filters with high time constants needed. This increases the occupied area of the solution on silicon. In addition, apart from the bank of filters, a conventional analog-to-digital converter (ADC) is required when the triggering event is detected. Therefore, system complexity and area are increased even more.

In this manuscript, we propose a "hybrid" approach for VADs applications that makes use of voltage-controlled-oscillators based ADCs (VCO-based ADCs) to perform the feature extraction of audio signals [10]. It is known that VCO-based ADCs are suitable for audio applications due to their low power consumption and dynamic range well-suited to human hearing [11–13]. VCOs can be also used to implement band-pass filters, such as in [14–16]. Here, we will make use of bi-quadratic filters implemented with VCO (specifically ring-oscillators) to extract the features of an audio input signal and generate a digital signal that could input a classification circuit. As the output of a ring-oscillator is a digital signal, the output of the filter will be a digital representation of the extracted features. Therefore in comparison with the analog approach (Figure 1b) we save power because we do not need the array of ADCs. The analog-to-digital conversion is already included in the features extractor stage and could be used when required to make a full analog-to-digital conversion of the input audio signal. Additionally, VCO-based filters do not require large capacitors for low-frequency band-pass structures. Finally, digital counters connected to the VCOs can be implemented with minimum-length transistors. In consequence, the proposed solution is expected to occupy a much lower area than the previous approaches, with less power consumption than the digital approach and competitive power consumption compared to the analog solution.

The document has the structure outlined below. Section 2 summarizes the conventional way of extracting the features with audio signals, paying particular attention to the Mel Frequency Cepstrum Coefficients (MFCCs) method. Section 3 theoretically shows the proposed VCO-based system, the behavioral model and the performance simulations made to validate it. In Section 4 the circuits designed for the implementation of the architecture are described. Making use of a 65-nm

CMOS process we are able to estimate the power consumption and the occupied area in a silicon implementation. Finally, Section 5 concludes the manuscript.

## 2. VAD Applications with Digital Custom Implementations

If we focus on VAD, applications we will deal with smart systems whose goal is being capable of detecting whether we are in the presence of human voice or not [17]. This supposes that some features must be necessarily extracted from the input sound stream and processed. To make this feature extraction the MFCCs are typically used in speech recognition and VAD tasks when working with the digital systems. Sounds generated by human voice are filtered out by the shape of the vocal tract. Knowing the shape of the vocal tract allows us to accurately define the representation of the phoneme which might be being produced. Additionally, this shape strongly depends on the power of the input stream sound. The purpose of calculating the MFCCs is to estimate this power, determine the presence of a phoneme and consequently distinguish between human voice and noise [18,19]. The conventional way of making this feature extraction is summarized in Figure 2. Once the input sound has been digitized, it is split up into frames. Then a pre-emphasis filter and windowing operations are applied to each frame. The FFT is used to calculate the spectrum of the frames and the result gets through a bank of Mel filters. Here the signal power within different frequency bands is estimated. After a log operation, the DCT is calculated to finally get the MFCCs. The MFCCs are the extracted features of the input signal that will feed a classifier circuit such as a neural network, a decision tree, etc. In the analog solutions proposed in the literature, an equivalent analog version of the MFCCs is extracted based on the energy split up into different bands of interest [8].

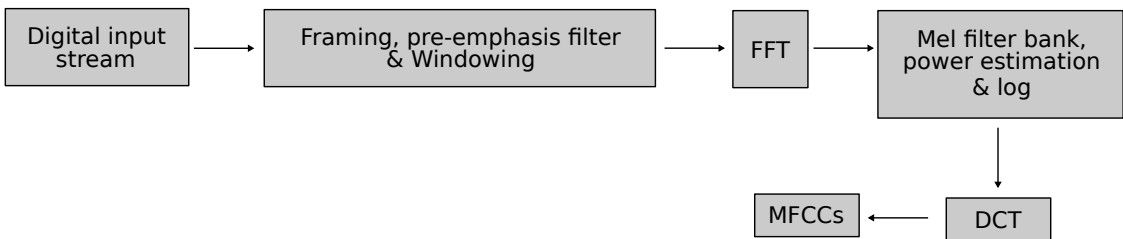

**Figure 2.** Mel Frequency Cepstrum Coefficient (MFCC) calculation flow.

## 3. VCO-Based Bi-Quadratic Filters

We propose a new way to extract the energy of an input audio signal within different frequency bands making use of highly scalable and ultra-low power VCOs occupying the minimum area. We will implement VCO-based integrators with ring-oscillators followed by digital counters [14,15]. This will allow us to build a bank of band-pass filters needed to extract features (energy within different frequency bands) in voice recognition tasks.

With that purpose in mind, we have selected the conventional bi-quadratic filter (Figure 3a). Our output is the band-pass filtered one, so that the Laplace-domain representation of the transfer-function will look as follows:

$$H(s) = \frac{-K \cdot \omega_o \cdot s}{s^2 + \frac{\omega_o}{Q}s + \omega_o^2},$$

(1)

where K is the input gain, Q is the quality factor and $\omega_o$ is the center frequency.

There exist architectures of low power and low area analog opamp-based bi-quadratic filters [20]. Nevertheless, the lower the center frequency the larger the capacitor and the resistor required. In consequence, the occupied area increases prohibitively.

### 3.1. Architecture for VCO-Based Bi-Quadratic Filter

The oscillation frequency of a VCO, $f_{\text{osc}}(t)$, with an input signal $x(t)$ follows:

$$f_{\text{osc}}(t) = f_o + K_{\text{VCO}} \cdot x(t), \quad x(t) \in [-1, 1], \tag{2}$$

where $f_o$ is the rest oscillation frequency and $K_{\text{VCO}}$ is the gain of the VCO. The input signal is assumed to be dimensionless and may vary between $-1$ and $1$.

According to [15], an integrator can be built with a pulse frequency modulator (PFM), composed of a VCO (that integrates the phase of the input signal), and an asynchronous digital counter (that quantifies the phase). Taking this equivalence into consideration, we can build a bi-quadratic filter with VCOs and counters just by replacing the conventional opamp-based integrators of Figure 3a with the equivalent VCO-based integrator structure. The resulting architecture is shown in Figure 3b.

Looking at one single VCO, for instance, $\text{VCO}_1$, we notice that we have a VCO connected to one "up" input of the counter and another VCO (reference VCO, $\text{VCO}_r$) connected to one "down" input of the counter. This is required to remove the phase offset of the $\text{VCO}_1$ and $\text{VCO}_2$ coming from the rest oscillation frequency $f_o$. The reference VCO always oscillates at $f_o$. Whereas the internal count value of the counters is increased by the rising edges in any "up" input, it is decreased by the rising edges in any "down" input. This way we quantize the phase of the VCOs and make the phase subtraction needed to implement the filter architecture. The result is digitally represented by the output of the counters.

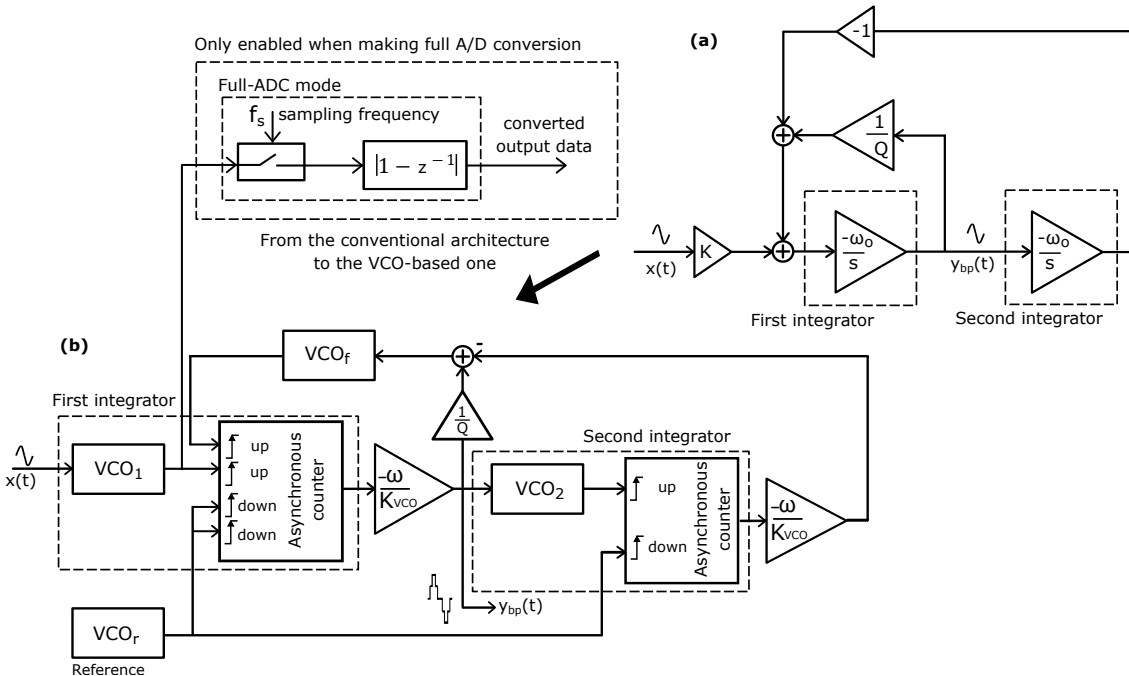

**Figure 3.** Conventional (**a**) and proposed (**b**) single-ended voltage-controlled-oscillators (VCO)-based architectures for a bi-quadratic filter.

Note that in the architecture depicted in Figure 3b the reference VCO ($\text{VCO}_r$) is required because there is no other way of removing the offset phase term. However, we will see later on that if we implement a pseudo-differential architecture this offset term can be canceled without any extra VCO, just by combining the outputs of the VCOs of both differential branches.

### 3.2. Behavioral Simulation

A behavioral model of the VCO-based system of Figure 3b was built and several simulations were made to validate the performance.

Firstly, to check the proper performance of the filter, an input signal composed of three sinusoidal waveforms spaced one decade between them in frequency was selected. The central input frequency was equal to 1 kHz. In relation to the VCO oscillation parameters, $f_o$ of 100 kHz and $K_{VCO}$ of 50 kHz were chosen for all the VCOs. The central frequency of the filter was selected to be 1 kHz as well. Figure 4 plots both the input signal (a) and the output signal (b) in time. The output signal (Figure 4b) has been low-pass filtered to avoid aliasing phenomena [11]. The output spectra of the input and the output signals are depicted in Figure 4c,d, respectively. As expected from (1), the most powerful component at the output corresponds to the closest input frequency to the center frequency of the filter, while the others are 20-dB attenuated.

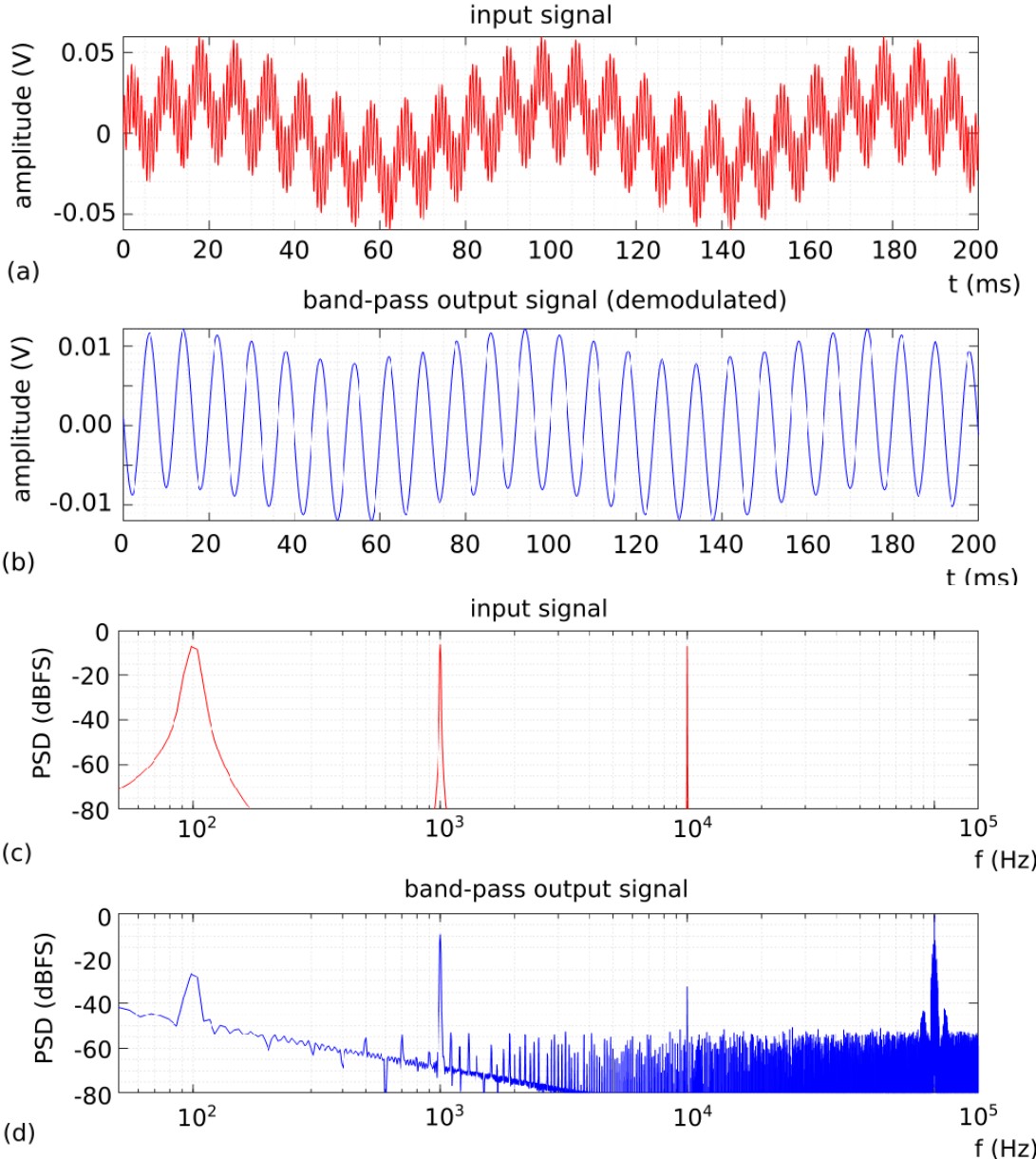

**Figure 4.** Behavioral simulation of the proposed VCO-based bi-quad filter: (**a**) input signal in time; (**b**) demodulated output signal in time; (**c**) input signal spectrum; and (**d**) output signal spectrum.

It is relevant to note that the filter output signal is actually a digital signal, corresponding to the output of a digital counter (Figure 3). This is of high relevance because we no longer require an ADC to digitize the analog extracted features. Furthermore, the first VCO (VCO$_1$) can be reused to make the

conventional analog-to-digital conversion after the detection stage. This strongly reduces the power consumption and the area in comparison to other speech recognition/VAD solutions, and confers on this solution an important advantage not present in other architectures.

The frequency response for a non-sinusoidal input (Figure 5a) wave has been also tested. Figure 5b depicts the output spectrum when the input signal is a sinc function with a bandwidth of 22 kHz. In Figure 5b, the red-colored spectrum depicts the output spectrum that would be obtained from the conventional bi-quadratic filter implemented with opamps, and the blue-colored one depicts the output spectrum obtained from our proposal. Whereas, for the band of interest the results are similar, in our proposal the sideband components due to the pulse frequency modulation can be appreciated at higher frequencies, similarly to VCO-based ADCs [11].

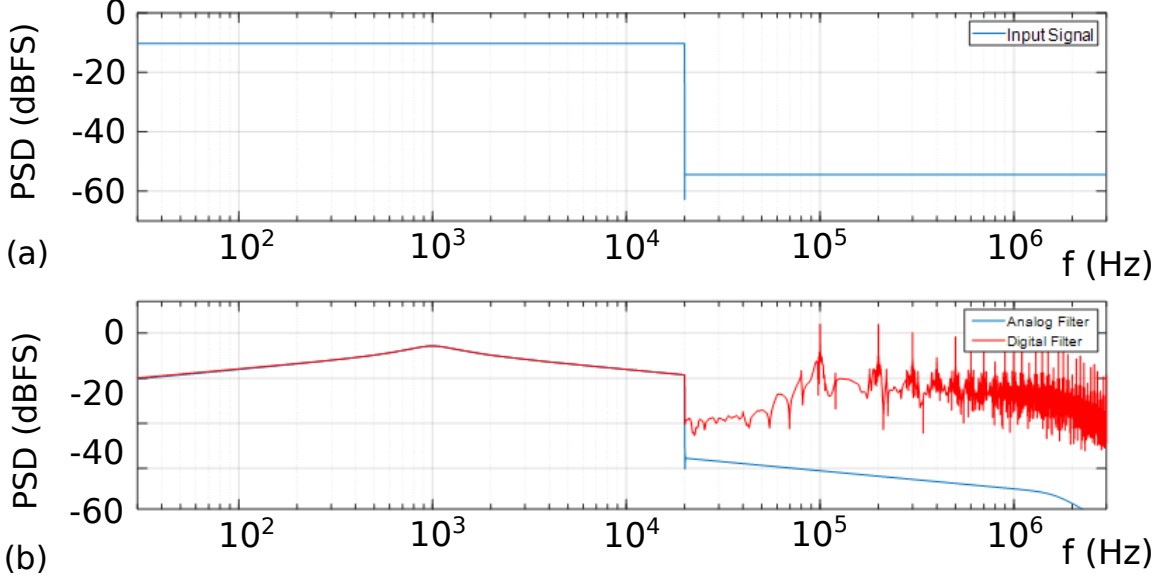

**Figure 5.** Behavioral simulation with sinc function as the input signal: (**a**) input signal spectrum; and (**b**) output signal spectrum (in blue the output spectrum with conventional bi-quad filter and in red the output spectrum with proposed VCO-based bi-quad filter).

Finally, one of the most common ways of generating the input signal for speech recognition/VAD applications consists of transforming the audio samples into spectrograms. The input signal is split up into frames of some ms (typically between 5 and 20 ms), and the energy within each frequency band is extracted. This way, if the whole system is composed of *M* different filters and the audio signal is divided into *N* frames, the input signal of the filter will become a *MxN* image. In Figure 6a we show an example of this. In this case, the input sample is a chirp function with an initial frequency of 20 Hz and a final frequency of 10 kHz. Low-frequency filters detect energy for low frequencies at the beginning and as the input frequency increases the power is shifted towards high-frequency filters. In Figure 6 each frame is of 5 ms, with 20 channels of band-pass filters. The center frequency increases 500 Hz along the bank of filters, starting at 20 Hz. For this example, the quality factor Q was kept constant (Q equals 3). That's the reason why the higher the center frequency, the higher the filter bandwidth and the higher the activity within different frequency bands [21]. The output data collected from Figure 6b could be the input of a decision stage for smart audio applications, with the advantages described above.

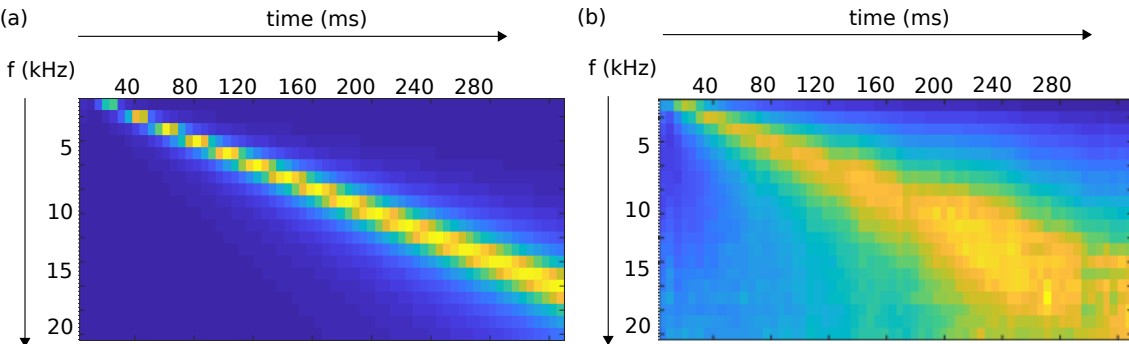

**Figure 6.** Spectrograms: (**a**) chirp function as input signal; and (**b**) output digital spectrogram.

### 3.3. Extension to Differential Configuration

As stated before, the architecture proposed in Figure 3b is a single-ended configuration where a reference VCO (VCO$_r$) is used to remove the phase offset [10]. If we extend this single-ended configuration to a pseudo-differential configuration, the reference VCO will become unnecessary. For the pseudo-differential configuration, we propose to make use of the architecture shown in Figure 7. The performance of this architecture was tested and the results were similar to the ones described in Section 3.2.

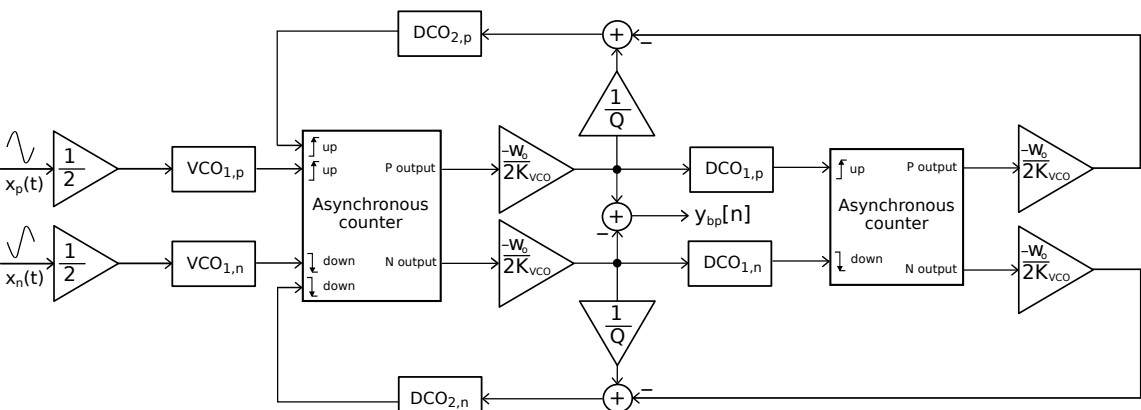

**Figure 7.** Pseudo-differential configuration of the VCO-based bi-quad filter.

## 4. Circuit Application

The architecture proposed in Figure 7 was designed in a 65-nm CMOS process to have an intuition about the area and power consumption we may expect from a silicon test. Although it is an old design process it is narrow enough to observe significant savings in power and area with respect to larger-length processes [10].

In this section, we will describe all the designed blocks, especially focusing on the VCOs and the asynchronous counters, and we will get an approach to the area and power consumption of each of them.

### 4.1. VCO and Front-End Circuit

The first element we need in a speech recognition task is a sound source. In this case, we chose a capacitor-based MEMS sensor in accordance with [12]. The performance of this sensor is based on the variation of a capacitance that depends on the input sound pressure. This capacitance, along with a biasing circuitry, generates a differential proportional voltage that is connected to the VCO$_{1,p}$ and VCO$_{1,n}$ of Figure 7. Thus, the resulting input stage will look as in Figure 8a.

The VCO is built with a ring-oscillator configuration [11], where the input voltage signal $x(t)$ is turned into a current $i(t)$ that feeds the inverters. In our proposal, these first ring-oscillators are composed of eleven taps. However, only one tap per ring-oscillator will be connected to the asynchronous counter afterwards (the P-side ring-oscillator to the "up" input and the N-side one to the "down" input, Figure 2b. The oscillation parameters of the VCO define the number of taps.

## 4.2. DCOs

If we look at the VCOs of Figure 7, we will notice that only the VCOs of the front-end will have an analog signal. The remaining ones will have a digital input signal, which means that they are digitally-controlled oscillators (DCOs). This supposes that they must be implemented with a different architecture with respect to the circuit of the ring-oscillator depicted in Figure 8. The designed circuit for the DCOs is shown in Figure 8b. Although the structure of the inverters connected in a ring configuration remains, the current that feeds them is digitally controlled by means of switches. These switches will be closed or not depending on the value of the input digital signal. As the weight of the digital inputs is the same for all of them, the input current will be mirrored from a reference current $I_0$. For the DCOs, an architecture with only three inverters is chosen.

## 4.3. Digital Logic Design: Asynchronous Counters

As depicted in Figure 7 the first counter is a 4-bit counter, with two inputs for the P-side and two inputs for the N-side. Figure 9a depicts the schematic of this counter. The four input signal gets into the clock input of four flip-flops and the outputs of these flip-flops are connected to a combinational digital logic that compares the count value for both sides of the differential configuration and makes the subtraction.

The second counter is a 2-bit counter with two inputs, one for each side of the differential configuration. Figure 9b shows the schematic of this counter. In this case, for the selected oscillation parameters, we only require a 2-bit counter, which significantly simplifies the circuit of the counter. For both counters, two very close edges with opposite directions (counting up and down) will not be lost due to the input flip-flops. If metastability occurs, the logic will only take a longer time to resolve.

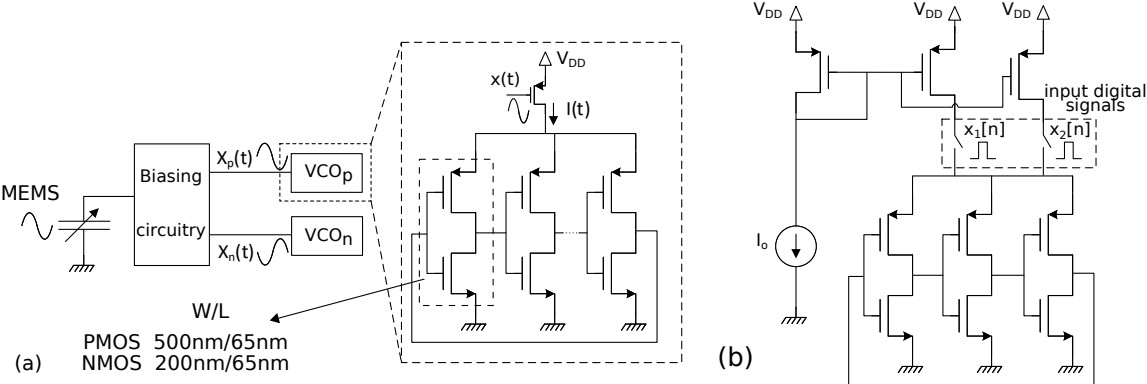

**Figure 8.** (**a**) Front-end circuit: MEMS, biasing circuitry and first VCOs; (**b**) three-tap digitally-controlled oscillator (DCO) circuit. The sizes for the transistors of the inverters are the same for (**a**) and (**b**).

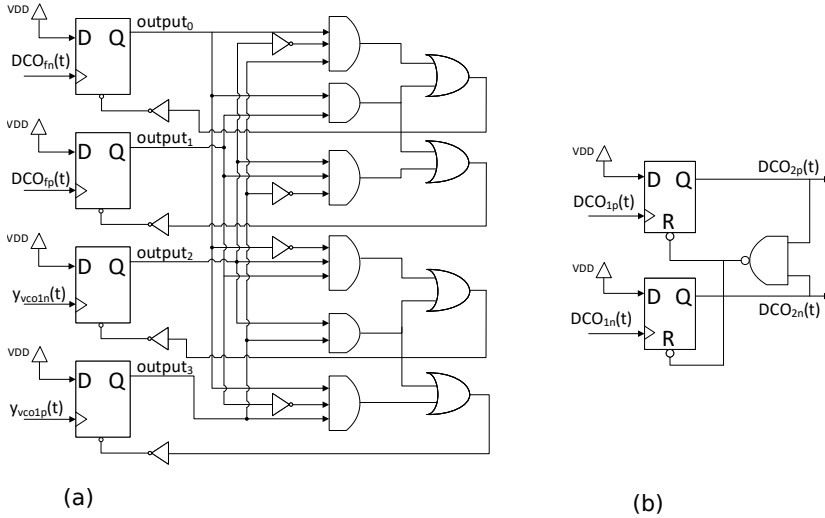

**Figure 9.** Asynchronous counters: (**a**) 4-bit counter; and (**b**) 2-bit counter.

### 4.4. Power Consumption and Area Estimation

To validate the proposed circuit, we have designed the previous circuits in a 65-nm CMOS process. The supply voltage for all of the described blocks is 1.0 V. The oscillation parameters for all the oscillators remain from the behavioral simulations of Section 2. With these conditions, we performed transient simulations to estimate the power consumption of the architecture. Regarding the ring-oscillators, we estimate the power consumption with an oscillation frequency that equals the rest oscillation frequency ($f_o$), which is the mean value of the input signal and is proven to provide accurate results. Regarding the counters, we calculate the average energy spent in an input transition by the counter and divide the result over $f_o$, for each of the inputs of the counters. We have added for these estimations the parasitic capacitances expected in layout in order to have more realistic results. Concerning the area, we take the area of the devices that compose each of the blocks and multiply it by three to consider the area needed for routing, guard-rings and pads. The results for the power consumption and the estimated area are shown in Table 1.

**Table 1.** Estimated power consumption and occupied area per channel, $V_{DD}$ = 1 V.

| Component | Current (nA) | Area ($\mu m^2$) | # | Total Current (nA) | Total Area ($\mu m^2$) | Total Power (nW) |
|---|---|---|---|---|---|---|
| VCO | 0.4 | 2 | 2 | 0.8 | 4 | 0.8 |
| DCO | 0.4 | 3 | 4 | 1.6 | 12 | 1.6 |
| 4-bit counter | 16.40 | 38 | 1 | 16.40 | 38 | 16.40 |
| 2-bit counter | 5.52 | 23 | 1 | 5.52 | 23 | 5.52 |
| **Total** | | | | **24.32** | **77** | **24.32** |

The values provided by Table 1 refer to one single channel of the feature extractor circuit. If we assume to make use of 20 channels to be able of making a proper decision, the whole power consumption will be equal to 0.48 $\mu$W and the occupied area will be 0.002 mm$^2$.

### 4.5. Circuit Impairments

The proposal of a whole architecture with the potential of being taped-out is not the scope of the manuscript. However, below we would like to make some considerations about circuit impairments we will have to face in case of going ahead with a prototype design.

The performance of ring-oscillators when included in an architecture such as the one shown in Figure 7 may vary because of several phenomena. Firstly, it is known that phase noise is the main limiting factor for high-resolution low-bandwidth applications, such as audio or biomedical

sensing. To overcome this issue noise simulations must be made first to calculate the input-referred phase noise [22] and compare it with the required resolution. Then, parameters like the number of phases, the power consumption and the devices' sizes must be adjusted to accomplish with that noise requirement. In our approach, phase noise will become of special relevance for the first ring-oscillator when making a full conversion once human voice has been detected. Strong noise requirements are not expected for the features extraction stage.

Secondly, open-loop VCO-based ADCs suffer from non-linearity behavior that generates distortion. This would be the case of the first ring-oscillator in Figure 7 which is out of the loop. Nevertheless, this is not a problem in audio applications because of the low amplitude of the input signal [13]. On the other hand, the purpose of using single-ended cells in the ring-oscillators is saving power. Although in the literature it is stated that this might suppose not enough PSRR performance, with a pseudo-differential architecture and a proper layout design, the PSRR will not become a limitation [12,13,23].

Finally, we propose an architecture composed of several ring-oscillators distributed over different channels but sharing the same silicon wafer. Undesired injection locking effects between different oscillators might occur. Thus a proper layout with a minimum distance between the different channels must be carried out at the expense of increasing the occupied area.

## 5. Comparison to State-of-the-Art Applications

In Table 2 our solution is compared to other equivalent solutions. Our proposal shows the best performance in terms of the area due to its high scalability and the mostly digital implementation. Additionally, the power consumption is competitive in comparison to other ultra-low-power analog solutions. Finally, it is the only solution in which the ADC is included in the features of extractor architecture. Note that the estimated area and power consumption only refers to the features extractor stage. In a complete system that makes both voice recognition and voice digitization, the first VCO of the architecture must accomplish with the conversion requirements required to make a full conversion of the analog input signal. Some extra digital logic is also needed (Figure 3b). Consequently, the results provided here will increase [12].

**Table 2.** Comparison to equivalent solutions.

|            | Power (μW) | Channel Number | Area (mm$^2$) | Approach      | ADC Included |
|------------|------------|----------------|---------------|---------------|--------------|
| This work  | 0.48       | 20             | 0.002         | Hybrid (VCOs) | Yes          |
| [6]        | 0.06       | 16–48          | 0.73 approx.  | Hybrid        | No           |
| [7]        | 6[1]       | 16             | 2             | Analog        | No           |
| [8]        | 0.38       | 16             | 0.16          | Analog        | No           |
| [9]        | >50        | -              | -             | Digital       | No           |

[1] The power of the classification stage is included.

## 6. Conclusions

A new approach for the implementation of the features extraction stage in speech recognition and VADs applications is described. Making use of VCO-based ADC filters we propose a hybrid solution between completely analog and completely digital architectures, leading to extraordinary area savings and competitive power consumption. The architecture is validated by behavioral simulations and designed in a 65-nm CMOS process. Estimations of the occupied area and the power consumption are made. The proposed solution is almost one hundred times smaller than the smallest architecture found in the literature, while keeping a similar power consumption than some of the analog solutions. Additionally, whereas the rest of the solutions need an ADC to digitize raw data when human voice has been detected, our solution could reuse part of the blocks included in the features extraction stage to perform the analog-to-digital conversion. This supposes a unique advantage of our solution with respect to the state-of-the-art. Finally, the solution is highly scalable due to the mostly digital nature of

the circuits, which means that higher power and area savings could be achieved if a narrower process is used. In comparison with [10], power consumption is reduced by 2.5 times and the occupied area is decreased more than 20 times.

**Author Contributions:** E.G., C.P., F.H, and L.H. are responsible for the architecture design. E.G., C.P., and F.H. wrote the manuscript. F.H. made the validation simulations and decided the system parameters. E.G, and C.P designed the architecture in the 65-nm CMOS process. L.H. managed, supported and supervised the research tasks, and proofread the manuscript. All authors have read and agreed to the published version of the manuscript.

**Funding:** This research was funded by the CICYT project TEC2017-82653-R, Spain.

**Conflicts of Interest:** The authors declare no conflict of interest.

## Abbreviations

The following abbreviations are used in this manuscript:

| | |
|---|---|
| ADC | Analog-to-Digital Converter |
| DCO | Digitally-Controlled-Oscillator |
| DCT | Discrete Cosine Transform |
| FFT | Fast Fourier Transform |
| MEMS | Microelectromechanical Systems |
| MFCC | Mel Frequency Cepstrum Coefficient |
| NMOS | N-type Metal Oxide Semiconductor |
| PFM | Pulse Frequency Modulator |
| PMOS | P-type Metal Oxide Semiconductor |
| PSRR | Power Supply Rejection Ratio |
| VAD | Voice Activity Detector |
| VCO | Voltage-Controlled-Oscillator |

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
