# Peer review of "Time-Encoding-Based Ultra-Low Power Features Extraction Circuit for Speech Recognition Tasks"

_electronics, doi:10.3390/electronics9030418_

Round 1

Reviewer 1 Report

Dear authors,

thank you for submitting a paper of good quality. I feel that in the table of comparison, a few typical VCO figures of merit would even improve the paper. I am particularly thinking of the VCO jitter and PSRR. Especially since you are using single ended inverters. Perhaps you could motivate this choice rather than fully-diff delay cells.

Best regards

Author Response

Please, you can find the answers in the attached .pdf file. Thanks.

Reviewer 2 Report

The paper proposes a new approach for circuits devoted to speech recognition using a bank a band-pass filters implemented with ring-oscillators to extract the features (energy within different frequency bands) of input audio signals and digitize them.

Comments:

  1. Improve the title. Currently, it is confusing. Avoid abbreviations.
  2. Why you selected 65-nm CMOS process technology to demonstrate application? This technology is 15 year old.
  3. The process of estimating power consumption should be described in more detail. Discuss various methods to perform such estimation, for example, at the behavioral level, see: DOI: 10.1155/2007/68673, or statistical method, see doi: 10.1007/s11277-018-5927-7
  4. Table 1 does not provide for power consumption.
  5. Check the numbering of Tables (Table II vs Table 2).
  6. Discuss the limitations of your approach and threats-to-validity of experimental results.
  7. Improve conclusions; support your claims with a summary of experimental results.

Author Response

Please, find the answers in the attached .pdf file. Thanks.

Round 2

Reviewer 2 Report

The authors have addressed all my remarks. I suggest to check carefully all figures and explain all abbreviations used (such as PMOS, NMOS) in respective figure captions.

Author Response

Thanks for your feedback.

We have checked all the figures and we have added the missing abbreviations to the list of abbreviations at the end of the document.